# Soft X-ray Tomography Reveals HSV-1-Induced Remodeling of Human B Cells

**DOI:** 10.3390/v14122651

**Published:** 2022-11-27

**Authors:** Jian-Hua Chen, Bieke Vanslembrouck, Axel Ekman, Vesa Aho, Carolyn A. Larabell, Mark A. Le Gros, Maija Vihinen-Ranta, Venera Weinhardt

**Affiliations:** 1Molecular Biophysics and Integrated Bioimaging Division, Lawrence Berkeley National Laboratory, Berkeley, CA 94720, USA; 2Department of Anatomy, University of California San Francisco, San Francisco, CA 94143, USA; 3Department of Biological and Environmental Science, Nanoscience Center, University of Jyvaskyla, 40014 Jyvaskyla, Finland; 4Centre for Organismal Studies, University of Heidelberg, 69120 Heidelberg, Germany

**Keywords:** X-ray tomography, soft X-rays, infection imaging, HSV-1, cell mapping, cryo imaging

## Abstract

Upon infection, viruses hijack the cell machinery and remodel host cell structures to utilize them for viral proliferation. Since viruses are about a thousand times smaller than their host cells, imaging virus-host interactions at high spatial resolution is like looking for a needle in a haystack. Scouting gross cellular changes with fluorescent microscopy is only possible for well-established viruses, where fluorescent tagging is developed. Soft X-ray tomography (SXT) offers 3D imaging of entire cells without the need for chemical fixation or labeling. Here, we use full-rotation SXT to visualize entire human B cells infected by the herpes simplex virus 1 (HSV-1). We have mapped the temporospatial remodeling of cells during the infection and observed changes in cellular structures, such as the presence of cytoplasmic stress granules and multivesicular structures, formation of nuclear virus-induced dense bodies, and aggregates of capsids. Our results demonstrate the power of SXT imaging for scouting virus-induced changes in infected cells and understanding the orchestration of virus-host remodeling quantitatively.

## 1. Introduction

Over the past few years, pathogenic infections have caused public health concerns, including huge impacts on the global economy. Those pathogens, as infectious agents, evade host defenses and remodel host cell machinery [1,2]. The remodeling of a host cell anatomy is a complex and dynamic process involving the rearrangement of the nucleus (DNA viruses) or intracellular membranes (RNA viruses) [3]. The difference in the scale of main players, i.e., from hundreds of nanometers for individual viruses to tens of micrometers for mammalian cells, and the intercellular communication of all organelles, makes it difficult to capture cellular rearrangement using merely one imaging technique.

Several recent studies have used super-resolution fluorescence microscopy [4] as well as volume electron microscopy (vEM) [5,6] to address the topics for understanding emerging viruses. Fluorescence microscopy offers superb temporal resolution for imaging interactions among labeled organelles in living cells without chemical- or cryo-fixations, while vEM offers better spatial resolution for imaging, allowing more detailed views of subcellular structures. However, the requirement of sophisticated sample preparations (fixation, labeling, and trimming) and time-consuming experimental protocols result in limited sample throughput [7,8,9].

Soft X-ray tomography (SXT) is increasingly used to image and quantify the subcellular structures in their most-native state without tagging with fluorescent markers or extensive sample preparations, such as sectioning [10,11]. SXT takes advantage of the X-ray “water-window”, where absorption of bio-organic structures prevails over water [12,13]. The quantitative measure of it, a linear absorption coefficient (LAC), is correlated with the bio-molecular density and concentration [14]. LAC is particularly informative for quantitatively investigating the reorganization and rearrangement of the subcellular organelles, such as chromatin compactions [15] and inactivation of chromosomes [16], and membrane reorganization during viral infection [17]. Furthermore, SXT has been successfully applied for research requiring higher biosafety levels, such as intracellular progress of reovirus in human cells [18] and highly infectious SARS-CoV-2 after the neutralization by aldehyde fixatives [19].

Currently, SXT imaging is possible using one of two alternative specimen holders: one for flat grids, mostly used for adherent cells, and one for thin-wall glass capillaries, mostly for imaging cells in suspension, including dissociated cells. While the use of flat grids allows for the correlation with existing fluorescence and electron microscopes [18,20,21], full-rotation tomography with capillaries as cell holders enables imaging of the entire cell with visualization of all cellular events [19]. Here, we have employed full-rotation SXT to capture the herpes simplex virus 1 (HSV-1) induced remodeling of human B cells.

HSV-1 infections are prevalent among humans and often asymptomatic. HSV-1 establishes latent infection in the sensory nervous system, which can result in recurrent infections due to the reactivation of the virus [22]. Previous research has shown that the structure of herpesvirus has a distinct four-layered structure: a core containing the double-stranded DNA (dsDNA) genome, enclosed by an icosahedral capsid, then surrounded by the tegument proteins, which are further encased in a glycoprotein-bearing lipid bilayer envelope. Electron tomography of HSV-1 demonstrated that the sphere-like enveloped virus particles are symmetrical with a diameter ranging from 155–240 nm [23]. The precise organization of the tegument and enveloped virus was studied by the combination of direct stochastic optical reconstruction microscopy (dSTORM) and stimulated emission-depletion (STED) microscopy, where the researchers proposed a model of protein organization inside the tegument with a diameter ranging from 125–245 nm [24,25].

The entry and egress of HSV-1 to the host cell is a complex procedure involving reorganizing and rearranging all cellular organelles. The development of morphological characteristics of herpesvirus infection proceeds in multiple stages and takes place both in the nucleus and cytoplasm. The viral capsid assembly, and DNA replication and packaging occur inside the host cell nucleus, while the viral protein production and tegument assembly take place within the cytoplasm [26].

Previous SXT experiments that focused on nuclear reorganization in viruses have established that formation of the viral replication compartment at late infection results in the enrichment of heterochromatin at the nuclear periphery accompanied by the compaction of chromatin [17]. Though compacted chromatin restricts viral capsid diffusion, SXT imaging showed the presence of interchromatin channels, which enable capsids to reach the nuclear envelope [27]. In addition, recent SXT studies on the remodeling of cytoplasm upon HSV-1 infection have shown active dynamics of cytoplasmic vesicles, lipid droplets, and elongation of mitochondria [28]. Thus far, spatiotemporal imaging of HSV-1 infections at the entire cell level over the course of infection has been limited.

In this work, we have used full-rotation SXT to visualize entire cells during HSV-1 infection by using a human B cell model. The high spatial resolution of 60 nm and the quantitative nature of soft X-ray absorption enabled us to resolve and quantitatively analyze internalized and secreted viral capsids. Intricate cytoplasmic structures in proximity to the nucleus could be visualized in 3D. Furthermore, based on the entire cell imaging, we could also capture rare events of presumably membrane-less condensates in the cytoplasm and nucleus.

## 2. Materials and Methods

### 2.1. Cell Culture and Viral Infection

The human B lymphocytes (GM12878, Epstein–Barr virus-transformed) were purchased from the NGIMS Human Genetics Cell Repository, Coriell Institute of Medical Research (Camden, NJ, USA). Human B cells were cultured and maintained at 37 °C, and 5% CO_2_ in RPMI-1640 medium supplemented with 15% of fetal bovine serum (FBS), L-glutamine, penicillin, and streptomycin (Gibco-Invitrogen, Carlsbad, CA, USA). Cells were sub-cultured every 2–3 days to maintain the density at ~1 × 10^6^ cells/mL. The EYFP-ICP4 HSV-1 strain, based on the 17+ strain, was a generous gift from R. Everett (MRC Virology Unit, Glasgow, UK). The viruses were amplified as previously described [29]. The cells were inoculated with HSV-1 EYFP-ICP4 at a multiplicity of infection (MOI) of 5 and kept at 37 °C until live-cell microscopy or fixation.

Human B cells infected with HSV-1 (EYFP-ICP4 tagged) were collected by centrifuging down (125× *g*, 10 min) at 12–16 and 20 h post-infection (hpi) into the pellet. The cell pellet was resuspended in Leibovitz’s L-15, phenol red-free culture medium (Gibco-Invitrogen) supplemented with 15% FBS (ATCC, Manassas, VA, USA). Fluorescence tagging (vEYFP-ICP4) was used as a gating marker for FACS sorting (BD FACS Aria, BD Biosciences, San Jose, CA, USA) to separate infected from non-infected cells. The cells were sorted into L-15 with 15% FBS growth medium to maintain viability, followed by cryo-fixation protocols. Cryo-fixation was done by fast plunging the loaded, cell-contained capillary sample holder into the liquid N_2_ cooled liquid propane tank; for a detailed description, see [30].

### 2.2. Soft X-ray Microscopy

To study the morphological changes of the subcellular ultrastructure during HSV-1 viral infection, we have imaged non-infected and infected human B cells in the soft X-ray microscope (XM-2) of the Advanced Light Source. The optical layout of the microscope end-station is shown in Figure 1. At XM-2, the illumination light source is produced by the 1.3 T bending magnet device and directed onto the condenser (KZP) by a flat nickel mirror. The beam focused by the KZP was order-sorted by the front pinhole onto the specimen and then magnified onto the CCD detector (ANDOR, model iKon-L DO936N BN9KH, 2048 × 2048 pixels) by another micro zone plate—MZP, see Figure 1b. Using switchable MZPs of 35 nm and 60 nm, determined by the outermost zone width (Weinhardt et al., 2020), we have matched the field of view of SXT with the size of the infected cells (field of view), see Figure 1c,d. A set of X-ray projection images were acquired over 180 degrees with a 2-degree rotation increment and exposure time of 200–500 ms, depending on the chosen magnification. The X-ray projections were then reconstructed by the SXT software into a 3D volume [31]. To visualize large cells, typically with signs of late-stage infection, we shifted cells vertically and acquired several datasets with an overlap of 20%, Figure 1d. These tomograms were then stitched by ImageJ [32] and Fiji [33] to analyze whole-cell anatomy.

### 2.3. Data Analysis

The automatic detection of virus particles was based on scale-space maxima of the Laplacian of the Gaussian. We defined candidates as local maxima of the response of size 1 pixel to 9 pixels and removed ones closer than 5 pixels to ensure that each fitted candidate contained only one particle.

A robust fitting scheme was used to separate well-defined particles. This was done by fitting an anisotropic gaussian function on the blob with two different cost functions, one minimizing the L2 norm of the fitted Gaussian function and one minimizing the background gradient, i.e., the image after subtracting a Gaussian function from the region of interest (ROI). Blobs were selected where the error between these two results (FWHM, full width at half maximum, and peak LAC) was less than 1%, and the peak contrast was over two standard deviations of the ROI.

Amira 2021.1 was used to manually annotate subcellular organelles. The criteria for identifying each organelle were described in previous publications [30,34]. In short, we calibrate the XM-2 optical system to deliver quantitation LAC values by use of polystyrene beads. The segmentation of cellular organelles is performed in two steps: (1) segmentation of cell cytoplasm and (2) segmentation of the nucleus. Both can be done by interpolating manually annotated ortho-slices. Organelles typically present in uninfected cells, such as lipid droplets, mitochondria, and ER, can be recognized from their representative LAC values. Organelles associated with viral infection are identified on differences in LAC, gross morphology, and comparative analysis with published structural analysis.

## 3. Results

SXT has been previously used to visualize the effect of HSV-1 on the marginalization of host chromatin [17,27] and elongation of mitochondria [28]. Though these effects can be seen in the current study, we have focused our results on highlighting the advantages of SXT, such as entire cell imaging, the quantitative nature of X-ray absorption contrast, and 3D subcellular structural information.

### 3.1. Remodelling of Entire Cells by HSV-1

The interaction of the HSV-1 virus with host cells upon infection is a complex process that involves the hijacking of multiple cellular organelles [35]. To map the orchestration of HSV-1 infection, we used full-rotation tomography and imaged entire human B cells susceptible to HSV-1 [27] without and with infection at 12–16 hpi and 20 hpi (Figure 2). In our previous studies, flow cytometry and real-time RT-PCR analyses and quantification of virus yield indicated that immediate-early early, late viral proteins, viral RNAs, and progeny viruses were produced in human B cells [27]. This suggests that HSV-1 could not only enter and infect B cells but also that its replication cycle was completed.

Based on the morphology and distinct LAC value in noninfected human B cells, we observed different subcellular structures, such as hetero- and euchromatin, the nucleoli, mitochondria, and lipid droplets (Figure 2a).

After 12–16 hpi, viruses were observed on the cell surface (Figure 2b). It was possible to detect them due to higher LAC values of 0.26 µm^−1^ compared to the absorption of cell filopodia, LAC of 0.20 µm^−1^. In a few cells (*n* = 1/7), we found large organelles of 1–2 µm in diameter with higher LAC, approximately 0.4–0.5 µm^−1^, compared to mitochondria’s LAC of 0.3 µm^−1^. Due to the lack of membranes in these large organelles, we suggest that they are stress granules, which are known signs of early HSV-1 infection [36]. These organelles contain multiple denser cores like those seen with fluorescence microscopy [37]. Later, at 20 hpi, cells were enlarged and structurally altered and seen with elongation of mitochondria, increased abundance of vesicles such as endosomes and lysosomes, and membrane-structure alterations, corresponding to the known hallmarks of HSV-1 infection [28,35]. Virus particles were observed on the cell surface; meanwhile, the nonenveloped capsids were detected inside the nucleus (Figure 2c, Appendix A). Based on the 2D line scan profile through two separated cell membrane-bound viruses, we estimated a full width at half maximum (FWHM) of ~180 nm diameter for individual capsids (Figure 2e). This size closely corresponds to enveloped HSV-1 virions with a range in diameter of 170–200 [23]. The FWHM measurement of nonenveloped capsid in the nucleus showed ~165 nm diameter (Figure 2f), which is relatively close to the capsid diameter of 125 nm measured for icosahedral capsids by electron cryomicroscopy [38].

### 3.2. Quantitative Analysis of HSV-1 Viral Particles

Upon line profile analysis of SXT data, we observed that LAC values of virus particles differ (Figure 2e,f). We then performed blob detection and statistical analysis of two major groups of virus particles: (i) on the cell surface and (ii) inside the nucleus. The LAC of viruses on the cell surface (*n* = 1429, from 3 infected cells) is 0.280 µm^−1^ (S.D. = 0.0488) with volume distribution (*n* = 1310, from 3 infected cells) of 6.71 × 10^−3^ µm^3^ (S.D. = 5.065 × 10^−3^), see Figure 3. Assuming that a virus is a spherical object, one can calculate that the diameter of a single virus on a cell surface is about 234 nm. In comparison, the viral particles detected inside the nuclei of the same cells (*n* = 2666, from 3 infected cells) have lower LAC of 0.238 µm^−1^ (S.D. = 0.0175). This increase in LAC by 1.18 corresponds to compositional differences of the enveloped viral capsids upon secretion from a cell versus non-enveloped viral capsids in the nucleus. The viruses assembled inside of the nucleus have a volume distribution (*n* = 2709, from 3 infected cells) of 2.60 × 10^−3^ µm^3^ (S.D. = 2.478 × 10^−3^ µm^3^). Assuming that the virus is a spherical object, one can calculate that the diameter of a single virus within the nucleus is about 170 nm. The physical sizes of nucleocapsids and virions on the cell surface measured from full rotation SXT align with the other study using grid-based SXT [28]. This suggests that SXT with either the capillary or 2D grid-based sample holder enables the separation of viral particles at various stages of their maturation.

### 3.3. Heterogeneity of Replication Compartments

In the nuclei of cells at 20 hpi, we observed periodic structures and foci that were not yet reported using SXT imaging (Figure 4). In the periphery of the nucleus, we noticed a condensed, highly absorbing membrane-less structure (Figure 4b,c, Appendix A). Upon further inspection, multiple condensed foci were identified with LAC > 0.32 µm^−1^. Such high LAC values are not typical for nuclear structures, where hetero- and euchromatin domains correspond to LAC of 0.32 µm^−1^ and 0.22 µm^−1^, respectively [15]. With the distinct LAC and proximity to nuclear protrusions, these structures are possibly nuclear inclusions of viral protein aggregates [39,40], viral protein quality control centers called VICE domains [41], or nuclear dense bodies [42].

In addition to those foci, multiple aggregates were resolved by SXT (Figure 4d). Based on 3D imaging from electron microscopy [43], these are most likely aggregations of HSV-1 capsids (Figure 4e). The periodicity of these aggregates is ~170 nm line width with a gross morphology of ~6.18 µm^3^, based on the segmented sub-area from SXT reconstruction (Figure 4f). Based on the 170 nm size of individual capsids inside the nucleus and periodicity of 170 nm, we estimate that there are approximately 2400 HSV-1 capsids in this measured structure.

### 3.4. Complex Three-Dimensional Membrane Structural Alterations

Enveloped viruses interact with cellular membranes using diverse strategies. Here we have tried to resolve the fine tubular structures associated with the HSV-1 virus [44].

We have observed the intricate cytoplasmic structures in cells at the late stage of infection, which include multi-layered cytoplasmic vesicular structures (Figure 5). During the viral egress from the nucleus to the cytoplasm, the fusion of the viral particles with the outer nuclear membrane involves membrane alterations to facilitate the viral translocation of the nucleocapsids to the cytoplasm. The mesoscale imaging reconstruction from SXT offers the opportunity to observe the cytoplasmic structural modifications during virus egress in 3D. We observed the intricate double membrane structures next to the nuclear membrane protrusion (Figure 5b,c) and the mitochondrial network (Figure 5c, Appendix A). In one of the structures, we found a concave membrane inclusion. The structures showed complex morphology which would not be visible in 2D imaging or section (Figure 5b,b’). These structures resemble multivesicular bodies, essential intermediates in the endolysosomal pathway. It has been suggested that multivesicular bodies most likely have a role in the HSV-1 gB protein-mediated envelopment of egressing viral capsids [45].

## 4. Discussion

To visualize and characterize cells and their subcellular reorganization, it is essential to understand the spatiotemporal dynamics of viral infections. In this study, we have used SXT to visualize the remodeling of whole cells induced by HSV-1. The ability to image an entire cell with full-rotation tomography with a high spatial resolution of 60 nm, and the quantitative nature of X-ray absorption contrast, makes SXT a unique imaging technique for capturing cell remodeling and rare events over the course of infections. Here, we have used SXT to capture HSV-1-infected human B cells, identify cellular alterations, show heterogeneity of viral replication compartments, and quantify viral particles.

The use of full-rotation tomography with soft X-rays provides several benefits: (1) capturing of complex orchestrated events with information on contact sites of different organelles [46]; (2) detection of rare events, like stress granules in this study, which are easy to miss with imaging techniques where only local structure is available; (3) ability to normalize quantitative parameters of organelles to cell and nucleus size [46,47,48] and (4) enabling robust and quantitative analysis of heterogenous cell population [49].

The linear absorption coefficient (LAC) from soft X-ray imaging offers a good way to accurately calculate the dimensions of the virus inside of the nucleus and on the cell membrane before and after their egress. Prior studies using fluorescence microscopes [25] and electron microscopes [23] have shown that the dimension of the virion varies depending on the modifications of the surface teguments along the virus production process. It remains challenging to precisely quantify the dimensions and bio-compositions using conventional imaging modalities in a statistically significant number of cells. In this study, we have demonstrated that rapid, automatic analysis of whole cells opens the door for measuring the accurate virion dimensions during egress with SXT. Our data is comparable to previous studies showing that the nucleocapsid diameter is about 170 nm [38]. We also show that the virus has significantly lower LAC values inside the nucleus when enveloped for secretion due to the additional tegument assembled in the cytoplasm before the cellular egress. The variation in the LAC values of the nuclear capsids can be explained by the existence of three capsid types (empty capsids, protein scaffold-containing capsids, and full nucleocapsids). These results demonstrate the sensitivity of the method to separate different maturation stages of virions.

Capsid aggregation in the nucleus of HSV-1-infected cells has been observed before using other methods [50]. Our data clearly shows a mixture of heterogeneous (in terms of LAC) membrane-less structures inside the nucleus. Even though SXT imaging cannot resolve individual virus capsids in these structures, our data—in combination with TEM data—estimate the total capsids in such periodic structures. This could be the basis for establishing a comprehensive model inside the nucleus that those compartments contribute to efficient viral replication, trafficking, and assembly. Combined with other tools, such as correlated fluorescence microscopy or biochemical analysis, this could shed light on detailed studies of host-pathogen interactions at the molecular level. The mesoscale SXT imaging also allows for direct observation of membrane alterations during the virus infection. By taking advantage of the full-rotation SXT tomographic reconstruction, the double membrane structures, as well as the nuclear membrane protrusions, can be seen in 3D without specific fluorescence labeling. SXT offers a convenient way of examining internal structures in a high-resolution manner.

Although access to a full-rotation SXT microscope is limited to one instrument at the moment, the development of laboratory SXT microscopes with the same capabilities [51,52,53] will pave the way to further high-throughput 3D imaging of entire cells in healthy and pathological conditions.

To summarize, we showed that high-resolution 3D imaging of entire cells made possible with full-rotation SXT captures cellular remodeling and rare events associated with HSV-1 infection in a human B cell model. We have captured signs of HSV-1 infection, including the formation of dense nuclear bodies and nuclear capsid aggregates. The spatial resolution and LAC sensitivity from SXT are sufficient to detect individual viral particles and discriminate different levels of virus particle maturation based on X-ray absorption. This quantitative information and the gross morphology of cells allow us to capture virus-host interactions at the cell population level. The isotropic 3D resolution of SXT enabled us to virtually dissect membrane budding in association with neighboring membrane alterations at the nuclear envelope. Quantitative analysis, such as precise 3D dimensions, organelle morphology, and the interactions among organelles enabled by full-rotation SXT, will facilitate understanding of virus-host interactions and the development of antiviral drugs.

## Figures and Tables

**Figure 1 viruses-14-02651-f001:**
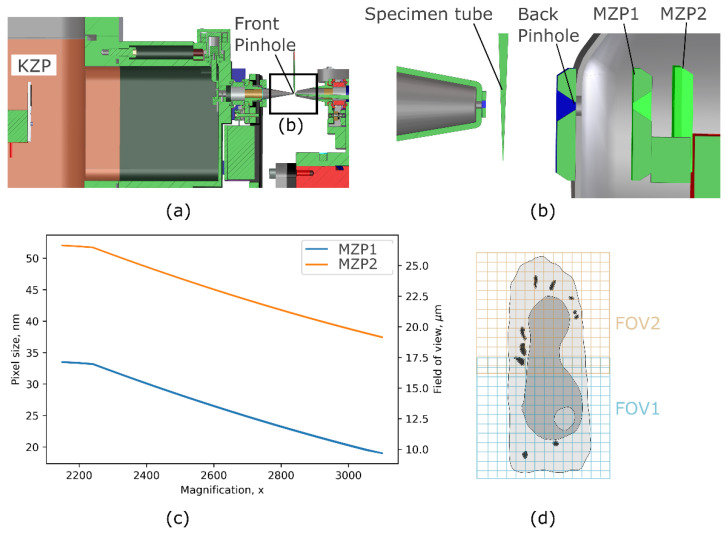
Schematics of soft X-ray microscope (XM-2). (**a**) Optical layout of the condenser zone plate (KZP), specimen tube, and objective zone plate MZP1 or MZP2 with 35 and 60 nm outermost zone width, respectively. (**b**) Magnification of specimen tube and multiple objectives installed at XM-2. (**c**) Pixel sizes in nm and the corresponding field of views in µm covered by two objectives in the SXT microscope with respect to magnification. (**d**) Acquisition strategy to overlay tomograms vertically for imaging larger cells inside the capillary.

**Figure 2 viruses-14-02651-f002:**
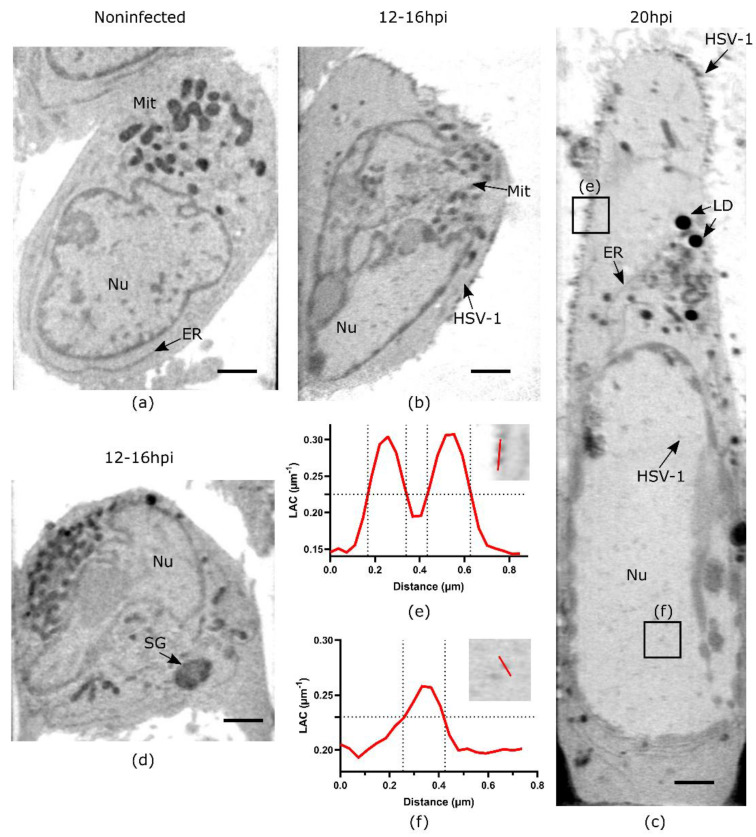
HSV-1 infection induced structural changes in human B cells. (**a**) Virtual slice through the non-infected cell. (**b**) Virtual slice through the cell at 12–16 hpi. (**c**) Virtual slice through the cell at 20 hpi. (**d**) Virtual slices through cells at 12–16 hpi demonstrate large organelles, presumably stress granules (SG). (**e**) Region of interest through enveloped capsids at cell membrane at 20 hpi and line profile showing size and X-ray absorption (LAC) values of SXT. (**f**) Region of interest through non-enveloped HSV-1 capsids inside the nucleus at late-stage infection and line profile showing size and X-ray absorption (LAC) values of SXT. Scale bars are 2 µm. All data are in greyscale from 0.2 to 0.5 µm^−1^ of SXT LAC values. Nu-nucleus, Mit-mitochondria, LD-lipid droplets, ER-endoplasmic reticulum, SG—stress granules.

**Figure 3 viruses-14-02651-f003:**
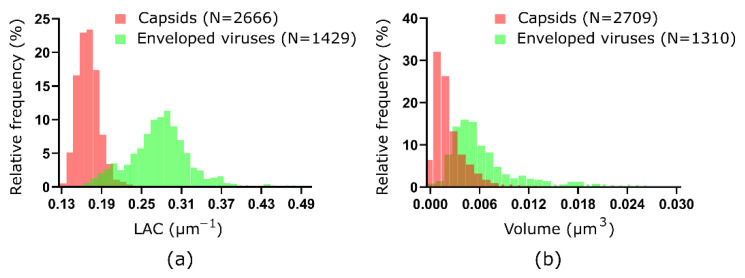
Quantitative analysis of nucleocapsids and secreted viruses of HSV-1. (**a**) Distribution of SXT LAC values for capsids (red) and enveloped viruses (green) based on blob detection. (**b**) Distribution of volume for capsids (red) and enveloped viruses (green) based on blob detection of the SXT volume.

**Figure 4 viruses-14-02651-f004:**
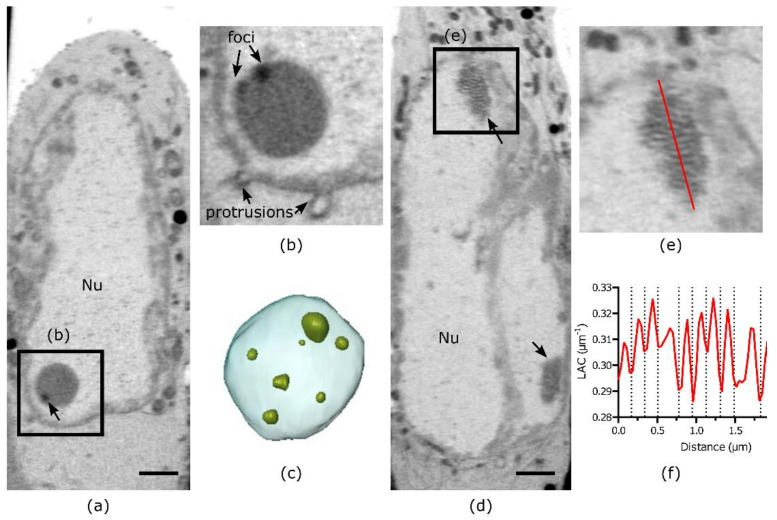
Structure of replication compartments inside HSV-1 infected cells. (**a**) Virtual slice through cell nucleus at late infection stage (20 hpi). (**b**) Region of interest from the same cell as in panel (**a**) highlighting multiple foci ((**a**,**b**) correspond to different z planes) and nuclear protrusions (**b**). (**c**) 3D rendering of the foci (green) inside the membrane-less compartment (light blue). (**d**) Virtual slice through cell nucleus at late infection stage (20 hpi) demonstrating two periodic structures (arrows). (**e**) Region of interest showing periodic structure from the cell in panel (**d**). (**f**) Line profile of LAC absorption values through the periodic structure as shown by red line in (**e**). Scale bars are 2 µm. All data is in greyscale from 0.2 to 0.5 µm^−1^ of SXT LAC values. Nu-nucleus.

**Figure 5 viruses-14-02651-f005:**
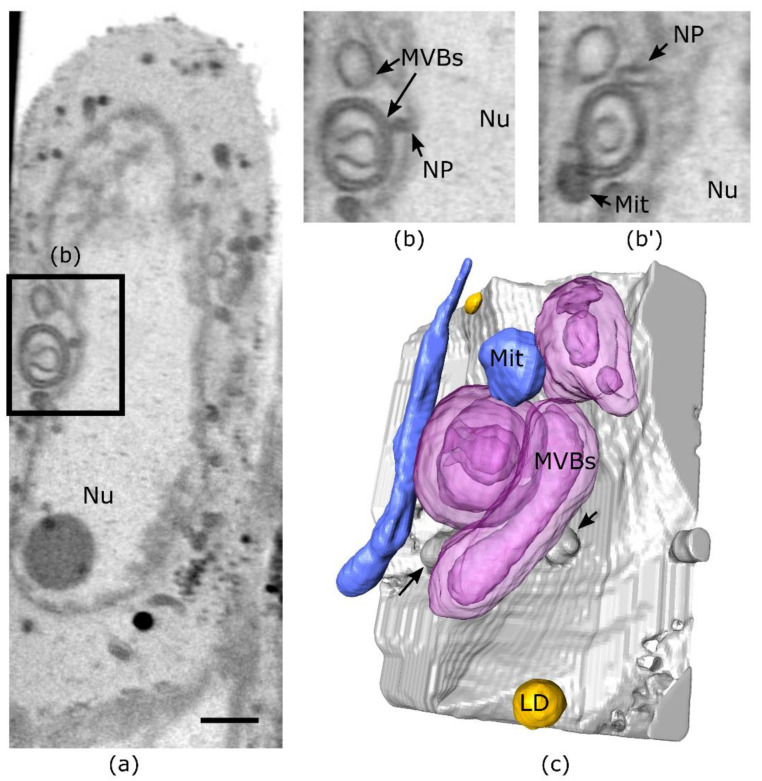
3D structure of membrane alterations induced by HSV-1. (**a**) Virtual slice through the cell at the late infection stage (20 hpi). (**b**) Region of interest from panel (**a**) highlighting double membrane structure with concave alteration that is not visible in the orthogonal plane and nuclear. (**b’**) Arrows show nuclear protrusions (NP). (**c**) 3D rendering of the multivesicular bodies (MVBs) with other organelles. Arrows point to nuclear protrusions. The scale bar is 2 µm. All data is in greyscale from 0.2 to 0.5 µm^−1^ of SXT LAC values. Nu-nucleus (gray), MVBs-multivesicular bodies (magenta), LD-lipid droplets (yellow), and Mit-mitochondria (blue).

## Data Availability

The data supporting this study’s findings are available from the corresponding author upon request.

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
