# Peer review of "Soft X-ray Tomography Reveals HSV-1-Induced Remodeling of Human B Cells"

_viruses, 2022, doi:10.3390/v14122651_

Round 1
Reviewer 1 Report
Chen and colleagues describe the use of soft x-ray tomography (SXT) to record in vitro cellular changes subsequent to infection of human B cells by herpes simplex virus type 1 (HSV-1). This well-written manuscript corroborates the work of others showing SXT as a means to assess the spatiotemporal dynamics of HSV-1 infection (See references 27/28). While aesthetically pleasing, observations reported in this manuscript are largely phenomenological or speculative and do not convincingly enhance our mechanistic understanding of viral infection or replication. The authors should consider expanding the manuscript and submitting as a review article.
Major concerns:
1. Considering the natural pathogenesis of HSV-1 infection, B cells are not prime targets. Accordingly, the broader applicability and translational validity of these observations are not abundantly clear.
2. With respect to the presumptive identification of viral stress granules (and other features), the authors note: “Further correlative fluorescence and SXT analysis is needed to confirm their identity” (lines 183-184). It is the opinion of this reviewer that other methods using specific labeling modalities must be implemented to provide validity to the observations herein using the same cellular source, MOI, and times post-infection.
Additional comments:
· The reference for the ICP4-eGFP reporter virus does not appear to match citation 29 that describes the use of tagged pseudorabies virus (PRV) from Lynn Enquist’s group, not Roger Everett. Please correct and clarify whether or not the ICP4-eGFP is also based on the strain 17 background. If a different strain is used, a comment should be included to justify the use given the appreciable differences in virulence among various strains of HSV-1.
· Given the high quality but general lack of novelty in this manuscript, the authors should consider expanding the work as a more comprehensive review of the topic at hand.
Author Response
Chen and colleagues describe the use of soft x-ray tomography (SXT) to record in vitro cellular changes subsequent to infection of human B cells by herpes simplex virus type 1 (HSV-1). This well-written manuscript corroborates the work of others showing SXT as a means to assess the spatiotemporal dynamics of HSV-1 infection (See references 27/28). While aesthetically pleasing, observations reported in this manuscript are largely phenomenological or speculative and do not convincingly enhance our mechanistic understanding of viral infection or replication. The authors should consider expanding the manuscript and submitting as a review article.
We appreciate your feedback. The manuscript was prepared or a special issue on novel imaging methos in viral research and therefore it’s scope is on the ability of SXT to visualize virus-host interactions.
Major concerns:
- Considering the natural pathogenesis of HSV-1 infection, B cells are not prime targets. Accordingly, the broader applicability and translational validity of these observations are not abundantly clear.
We have incorporated reasons why we chose B cells in our studies and references supporting found intracellular changes.
- With respect to the presumptive identification of viral stress granules (and other features), the authors note: “Further correlative fluorescence and SXT analysis is needed to confirm their identity” (lines 183-184). It is the opinion of this reviewer that other methods using specific labeling modalities must be implemented to provide validity to the observations herein using the same cellular source, MOI, and times post-infection.
We agree with the Reviewer that it is always good to use several imaging techniques combined with advanced analyses to identify cellular structures. In our previous studies, we analyzed the HSV-1 infected B cells with the same MOI and infection times by using EM, laser scanning microscopy of fixed and living cells (Myllys 2016, Aho 2017). We have decided to focus this manuscript on the novel potential of SXT imaging techniques in infected cells. We thus removed the statement on correlation and toned down conclusions on organelles.
Additional comments:
The reference for the ICP4-eGFP reporter virus does not appear to match citation 29 that describes the use of tagged pseudorabies virus (PRV) from Lynn Enquist’s group, not Roger Everett. Please correct and clarify whether or not the ICP4-eGFP is also based on the strain 17 background. If a different strain is used, a comment should be included to justify the use given the appreciable differences in virulence among various strains of HSV-1.
The reference has been replaced by Everett et al. J. Virol. 2003, 77, 3680–3689. We have clarified the origin EYFP-ICP4 strain by adding “which is based on the 17+ strain”(Materials and methods, page 3, lines 112-114).
Given the high quality but general lack of novelty in this manuscript, the authors should consider expanding the work as a more comprehensive review of the topic at hand.
Thank you for this suggestion. Many aspects of our work were not reported previously, we therefore believe our manuscript stands as original article.
Reviewer 2 Report
The manuscript entitled "Revealing the effect of HSV-1 Virus in Host Crlls by Soft X-ray Tomography" is attractive to the readers. The auhors used full-rotation SXT to visualize human B cells infected by HSV-1 and applied the metrology of linear absorption cofficient (LAC) to segment each organelle from HSV-1 infected B cells.
Questions:
1. The characters in Figure 1 (a) and (b) are not clear, especially the numbers in each red rectangle indicating the distance, please revise it.
2. Please check again in the text section 3.3 from line 239 to 245. The figure should be Figure 4 (b), 4(e) and 4 (f) or should be Figure 3 (b), 3 (e) and 3 (f) ?
3. Please calculate again in the text section 3.3 from line 239 to 245, If the volume of aggregation of HSV-1 capsides is about 6.18 um3, and the size of individual capsides inside the nucleus is 170 nm, Will the number of capsides be 300 or 2400 ?
4. The authors applied LAC to analysis the organelles. I would recommend authors to give a brief description of LAC in data analysis. From LAC analysis without using the dyeing, how do the authors distinguish the organelle that is a stress granule (SG) or a damaged mitochondria in Figure 2(d) ?
Author Response
The manuscript entitled "Revealing the effect of HSV-1 Virus in Host Cells by Soft X-ray Tomography" is attractive to readers. The authors used full-rotation SXT to visualize human B cells infected by HSV-1 and applied the metrology of linear absorption coefficient (LAC) to segment each organelle from HSV-1 infected B cells.
Questions:
- The characters in Figure 1 (a) and (b) are not clear, especially the numbers in each red rectangle indicating the distance, please revise it.
The image has been modified as suggested
- Please check again in the text section 3.3 from line 239 to 245. The figure should be Figure 4 (b), 4(e) and 4 (f) or should be Figure 3 (b), 3 (e) and 3 (f) ?
The text and figure legend concerning Figure 4 has been checked and modified.
- Please calculate again in the text section 3.3 from line 239 to 245, If the volume of aggregation of HSV-1 capsids is about 6.18 um3, and the size of individual capsids inside the nucleus is 170 nm, Will the number of capsids be 300 or 2400?
Thanks for pointing out of this calculation. With 170nm diameter of the individual capsids inside the nucleus, there should be ~2400 HSV-1 capsids in this measured structure.
6.18/(2.6x10-3)=2376
- The authors applied LAC to analysis the organelles. I would recommend authors to give a brief description of LAC in data analysis. From LAC analysis without using the dyeing, how do the authors distinguish the organelle that is a stress granule (SG) or a damaged mitochondria in Figure 2(d) ?
We have extended description of LAC analysis in the Data Analysis chapter 2.3.
Reviewer 3 Report
In this paper, the authors used soft x-ray tomography (SXT) to measure the 3D structures in human B cells infected with herpes simplex virus (HSV). This is the first such report using this technique in this setting, and as such is largely proof-of-concept rather than focusing on a specific finding. While lymphocytes aren't typically infected by HSV in vivo, their nature as suspension cells allow full-rotation tomography for greater detail.
The data for virion size matches expected values from studies using different techniques, however the measurement of linear absorption coefficient (LAC) gives a better window for measuring diversity of virions than volume estimation alone. They were also able to image several other bodies, such as (what is likely) a VICE domain, replication compartments, and a curious membraned vesicle, whose properties would not be fully describable with sectional techniques. Such techniques will be useful in following up on the structures described.
Overall, the authors have established SXT as a viable technique in this setting. The paper is well written, but there are a few minor text edits that might improve things:
Figure 1B: change the um to micrometer for the focal length, in the red box put a space in andthe and center "nickel core"
Line 267: add a comma after DMVs
Line 268: add a comma after morphology
Line 279: add a comma after reorganization
Lines 339-340: there is no supplemental information, so this section can be removed
Author Response
In this paper, the authors used soft x-ray tomography (SXT) to measure the 3D structures in human B cells infected with the herpes simplex virus (HSV). This is the first such report using this technique in this setting, and as such is largely proof-of-concept rather than focusing on a specific finding. While lymphocytes aren't typically infected by HSV in vivo, their nature as suspension cells allows full-rotation tomography for greater detail.
The data for virion size matches expected values from studies using different techniques, however, the measurement of linear absorption coefficient (LAC) gives a better window for measuring the diversity of virions than volume estimation alone. They were also able to image several other bodies, such as (what is likely) a VICE domain, replication compartments, and a curious membraned vesicle, whose properties would not be fully describable with sectional techniques. Such techniques will be useful in following up on the structures described.
We appreciate your positive feedback and have implemented all suggestions.
Overall, the authors have established SXT as a viable technique in this setting. The paper is well written, but there are a few minor text edits that might improve things:
Figure 1B: change the um to micrometer for the focal length, in the red box put a space in and the and center "nickel core"
We have modified the figure leaving only information described in the manuscript.
Line 267: add a comma after DMVs
This sentence has been changed and a comma is not necessary anymore.
Line 268: add a comma after the morphology
The comma has been added
Line 279: add a comma after reorganization
The comma has been added
Lines 339-340: there is no supplemental information, so this section can be removed
We have extended supplementary information with Video files.
Reviewer 4 Report
In this manuscript by Chen et al., the authors use the technique of "full-rotation" soft X-ray tomography (SXT) to examine the effects of herpes simplex virus type-1 (HSV-1) infection on the host cell as it replicates. This is a fairly novel approach that has not been used extensively to examine HSV-1-infected cells. The authors do present some interesting findings, but overall I feel that several aspects of this paper need considerable more work. Below are my comments in point form.
1. line 89: "Thus far, spatiotemporal imaging of HSV-1 infections at entire cell level over the course of infection has not been performed by SXT". This is an arguable statement in my opinion as a similar paper was recently (2022) published in PLoS Pathogens: Nahas et al., "Near-native state imaging by cryo-soft-X-ray tomography reveals remodelling of multiple cellular organelles during HSV-1 infection" (PMID: 35797345). It could be that the distinction in the authors' minds is that Chen et al. used "full-rotation" SXT, whereas the Nahas et al. study used adherent cells and flat grids. If so, perhaps the authors could flesh out this argument a bit. As it is, it seems as if the authors are "lowballing" the Nahas et al. paper, which they do reference, although somewhat obliquely.
2. One area in particular where the authors should discuss their results in comparison to those of Nahas et al. is in the experiments shown in Fig. 3. Here, the authors use SXT to compare the physical size of nucleocapsids inside the cell to that of virions on the cell surface. The data fit nicely with the expected size of capsids vs. mature particles, and thus show that this technique can be used to follow virion maturation. But Nahas et al. appear to have done similar experiments and got similar results, as shown in in Fig. 1 of their PLoS Pathogens paper. Shouldn't their work be cited and discussed?
3. The choice of a human B lymphocyte cell line by the authors makes some sense because the authors presumably need to use non-adherent host cells for "full-rotation" SXT. But a lymphocyte is a somewhat unusual host cell for an HSV-1 study. Given this, do the authors know whether HSV-1 replicates productively in these cells, and with similar kinetics to adherent host cells? With regard to time points, using 12-16 hpi as the "early" (see Fig. 2 legend) time-point for a high MOI infection seems odd, as this would be "late" in a epithelial cell infection.
4. As shown in Fig. 2D, the authors visualize membraneless cytoplasmic bodies in infected cells which they "suggest" are stress granules, based on their characteristics (line 181). They admit that more "analysis is needed to confirm their identity". Despite this uncertainty, they go on to label the bodies as "SG" in Fig. 1D. Furthermore, they state in the Abstract (line 22) that they "observed" stress granules, and in the Discussion that they "detected" stress granules (line 289). The authors should dial down the conclusion unless there is a way to provide further evidence for stress granules.
5. The authors argue that SXT is a good technique to capture "rare events" (Abstract, line 21). This may be true, although it could be argued that is the common events are the most interest. In any case, the frequency of observed events is an issue in the results shown Fig. 4A-C. Here, the authors identify foci-containing membraneless nuclear structures and show that these bodies are often near nuclear protrusions. This is an intriguing finding. But we need some information on the frequency of these structures, e.g. what percentage of cells have them, and how many are there per cell? Is the formation of these bodies a rare or common event.
6. The most important unknown in regard to the nuclear bodies shown in Fig. 4A-C is what they are. Here, the authors' discussion is quite confusing, and I came away with little idea of what these structures might be. On line 235, they speculate that "these structures could be the protein quality control center during HSV-1 infection". Since they don't provide a reference, it is unclear what they are referring to, but it has been proposed that structures called "VICE domains" serve a protein quality control function in HSV-1 infection (Livingston et al., 2009, PMID 19816571). In the very next sentence, they go on to suggest that the "sub-domains are regions with function similar to nuclear inclusion bodies, known players in HSV-1 infection." The reference here is to a paper that describes ND10 domains/PML bodies, which are distinct structures from VICE domains. However, these seem unlikely to be present since the HSV-1 ICP0 protein usually disperses these bodies early in infection. Surprisingly, the the authors don't mention the possibility that the bodies are "replication compartments", which are certainly the best characterized of the virus-induced bodies seen in HSV-1-infected cells (for recent review, see Knipe et al., Ann. Rev. Virol. 9:307-327, 2022). It also seems possible that the bodies haven't been observed before - perhaps they are unique to lymphocyte infections? Overall, more work is needed to sort this out.
Minor issues:
- Title "HSV-1 Virus" is redundant since "HSV" stands for "herpes simplex virus".
- line 79: "DNA replication and packing" should be "DNA replication and packaging".
- line 81: " tegument assembly and acquiring take place within the cytoplasm". Acquiring of what?
- line 106: " The cells were inoculated with wild-type HSV-1 17+ or HSV-1 EYFP-ICP4". It should be spelled out which experiments were done with which virus. Also is HSV-1 EYFP a phenotypically wild-type virus?
- line 289, Discussion: The authors state that SXT is good for "rare events, like stress granules". I don't think one can call a stress granule an "event". There are similar issues in the Abstract.
Author Response
In this manuscript by Chen et al., the authors use the technique of "full-rotation" soft X-ray tomography (SXT) to examine the effects of herpes simplex virus type-1 (HSV-1) infection on the host cell as it replicates. This is a fairly novel approach that has not been used extensively to examine HSV-1-infected cells. The authors do present some interesting findings, but overall I feel that several aspects of this paper need considerably more work. Below are my comments in point form.
Thank you for your positive response, we addressed all comments.
- line 89: "Thus far, spatiotemporal imaging of HSV-1 infections at the entire cell level over the course of infection has not been performed by SXT". This is an arguable statement in my opinion as a similar paper was recently (2022) published in PLoS Pathogens: Nahas et al., "Near-native state imaging by cryo-soft-X-ray tomography reveal remodeling of multiple cellular organelles during HSV-1 infection" (PMID: 35797345). It could be that the distinction in the authors' minds is that Chen et al. used "full-rotation" SXT, whereas the Nahas et al. study used adherent cells and flat grids. If so, perhaps the authors could flesh out this argument a bit. As it is, it seems as if the authors are "lowballing" the Nahas et al. paper, which they do reference, although somewhat obliquely.
Our apologies, we have not intended to undermine work of Nahas et al. This work actually inspired us to look at the multiple cytoplasmic changes and we cite it multiple times.
We have tuned the introductory statement to “Thus far, spatiotemporal imaging of HSV-1 infections at the entire cell level over the course of infection has been limited.” And revised other sections of the manuscript accordingly.
- One area in particular where the authors should discuss their results in comparison to those of Nahas et al. is in the experiments shown in Fig. 3. Here, the authors use SXT to compare the physical size of nucleocapsids inside the cell to that of virions on the cell surface. The data fit nicely with the expected size of capsids vs. mature particles and thus show that this technique can be used to follow virion maturation. But Nahas et al. appear to have done similar experiments and got similar results, as shown in Fig. 1 of their PLoS Pathogens paper. Shouldn't their work be cited and discussed?
We have added statement on similar results in citations on page 7 line 224.
- The choice of a human B lymphocyte cell line by the authors makes some sense because the authors presumably need to use non-adherent host cells for "full-rotation" SXT. But a lymphocyte is a somewhat unusual host cell for an HSV-1 study. Given this, do the authors know whether HSV-1 replicates productively in these cells, and with similar kinetics to adherent host cells? With regard to time points, using 12-16 hpi as the "early" (see Fig. 2 legend) time point for a high MOI infection seems odd, as this would be "late" in an epithelial cell infection.
For soft X-ray tomography (SXT) image acquisition, the cells have to be placed in (cylindrical) thin-walled capillaries of a diameter of up to 15 µm. B cells were used in these studies due to their small size and HSV-1 susceptibility. In our earlier studies flow cytometry and real-time RT-PCR analyses together with quantification of the virus yield indicated that immediate-early, early, and late viral proteins, viral RNAs, and progeny viruses were produced in B cells (Myllys 2016). This suggests that HSV-1 not only could enter and infect B cells but that its replication cycle was completed.
We have added corresponding explanations on page 4 line 176-180.
We agree that 12-16 hpi is definitely not an early infection. The text has been modified by replacing ”early infection stage” with ”12-16 hpi”.
Ref.
Myllys et al. Herpes simplex virus 1 induces egress channels through marginalized host chromatin. Sci Rep. 2016 Jun 28;6:28844.
https://pubmed.ncbi.nlm.nih.gov/27349677/
- As shown in Fig. 2D, the authors visualize membraneless cytoplasmic bodies in infected cells which they "suggest" are stress granules, based on their characteristics (line 181). They admit that more "analysis is needed to confirm their identity". Despite this uncertainty, they go on to label the bodies as "SG" in Fig. 1D. Furthermore, they state in the Abstract (line 22) that they "observed" stress granules, and in the Discussion that they "detected" stress granules (line 289). The authors should dial down the conclusion unless there is a way to provide further evidence for stress granules.
We have toned down all statements on stress granules throughout the manuscript.
- The authors argue that SXT is a good technique to capture "rare events" (Abstract, line 21). This may be true, although it could be argued that the common events are the most interesting.
The abstract has been modified and the sentence containing “rare events“ has been replaced.
In any case, the frequency of observed events is an issue in the results shown in Fig. 4A-C. Here, the authors identify foci-containing membraneless nuclear structures and show that these bodies are often near nuclear protrusions. This is an intriguing finding. But we need some information on the frequency of these structures, e.g. what percentage of cells have them, and how many are there per cell? Is the formation of these bodies a rare or common event?
We revisited the literature on HSV-1-induced nuclear changes and we noticed that these dense nuclear structures are most likely earlier described HSV-1 infection-induced dense nuclear bodies (Besse 1996). The bodies contain viral proteins (e.g. UL3, UL4), nucleolar proteins nucleolin, and fibrillarin (Jahedi 1990, Markovitz 2000), however, their biological function is still unknown. The whole manuscript has been modified and the emergence of infection-induced dense nuclear bodies has been included. As shown by previous studies mentioned, and in our own EM, and confocal microscopy observations (projects in the process), dense bodies (one or sometimes several) are frequently seen in the nucleus of infected cells at the late stage of infection.
Besse S, Puvion-Dutilleul F. Intranuclear retention of ribosomal RNAs in response to herpes simplex virus type 1 infection. J Cell Sci. 1996 Jan;109 ( Pt 1):119-29. https://pubmed.ncbi.nlm.nih.gov/8834797/
- The most important unknown in regard to the nuclear bodies shown in Fig. 4A-C is what they are. Here, the authors' discussion is quite confusing, and I came away with little idea of what these structures might be. On line 235, they speculate that "these structures could be the protein quality control center during HSV-1 infection". Since they don't provide a reference, it is unclear what they are referring to, but it has been proposed that structures called "VICE domains" serve a protein quality control function in HSV-1 infection (Livingston et al., 2009, PMID 19816571). In the very next sentence, they go on to suggest that the "sub-domains are regions with function similar to nuclear inclusion bodies, known players in HSV-1 infection." The reference here is to a paper that describes ND10 domains/PML bodies, which are distinct structures from VICE domains. However, these seem unlikely to be present since the HSV-1 ICP0 protein usually disperses these bodies early in infection. Surprisingly, the authors don't mention the possibility that the bodies are "replication compartments", which are certainly the best characterized of the virus-induced bodies seen in HSV-1-infected cells (for a recent review, see Knipe et al., Ann. Rev. Virol. 9:307-327, 2022). It also seems possible that the bodies haven't been observed before - perhaps they are unique to lymphocyte infections? Overall, more work is needed to sort this out.
Minor issues:
- Title "HSV-1 Virus" is redundant since "HSV" stands for "herpes simplex virus".
True, the title has been corrected
- line 79: "DNA replication and packing" should be "DNA replication and packaging".
Yes, this was corrected
- line 81: " tegument assembly and acquiring take place within the cytoplasm". Acquiring of what?
The sentence cleared and the word acquiring was removed
- line 106: " The cells were inoculated with wild-type HSV-1 17+ or HSV-1 EYFP-ICP4". It should be spelled out which experiments were done with which virus. Also, is HSV-1 EYFP a phenotypically wild-type virus?
We have corrected the usage of EYFP-ICP4 virus only. The EYFP-ICP4 virus is based on the wt 17+ strain, and in our experiments, we do not see differences in the progression of infection between them in the parallel control studies.
- line 289, Discussion: The authors state that SXT is good for "rare events, like stress granules". I don't think one can call a stress granule an "event". There are similar issues in the Abstract.
The abstract and discussion have been remodified and events have been removed from both.
Round 2
Reviewer 1 Report
Chen and colleagues describe the use of soft x-ray tomography (SXT) to record in vitro cellular changes subsequent to infection of human B cells by herpes simplex virus type 1 (HSV-1). This well-written manuscript corroborates the work of others showing SXT as a means to assess the spatiotemporal dynamics of HSV-1 infection (See references 27/28).
This reviewer’s overall opinion of this work remains unchanged. While aesthetically pleasing, observations reported in this manuscript are largely phenomenological or speculative and do not convincingly enhance our mechanistic understanding of viral infection or replication. That said, the editors might feel that this manuscript fits the scope of the special topic issue on novel imaging modalities in viral research.
The cell model utilized (i.e. B cells) is indisputably withheld from the title, abstract, introduction, and discussion. This approach seems potentially misleading. The authors’ previous work (Ref. 27; Myllys et al. https://pubmed.ncbi.nlm.nih.gov/27349677/) establishes the infectivity of HSV-1 strain 17 on B cells in vitro. However, supplemental data from that manuscript clearly demonstrates that B cell infection is uncommon (<10%) even at a high multiplicity of infection (MOI = 5). This represents a critical difference between B cells and susceptible cells (e.g. epithelial cells). Accordingly, the authors do not provide demonstrable evidence that the phenomena observed in vitro via SXT in infected B cells have any bearing on susceptible cells relevant to the natural pathogenesis of HSV-1. The unorthodox cell model and concomitant major research limitation must be clearly highlighted prior to publication, even if the primary focus is on the potential of SXT as an imaging tool for viral research.
Author Response
We apologize for the confusion and we agree that we should have more clearly pointed out that our manuscript represents a successful application of Soft X-Ray Tomography in a cellular model using HSV-1 infected human B cells. Appreciating the Reviewer’s suggestion, we have modified the title, abstract, introduction, and Discussion to clarify this. Please see the corrected and marked manuscript.
Reviewer 4 Report
I previously reviewed this manuscript and had several criticisms. However, the authors have thoughtfully taken those criticisms into account and done an excellent job of revising the paper. I now think that the paper is in very good shape and that it will be of interest to those working on HSV-1 as well as those interested in imaging techniques to study viral infections. I have only two minor issues for the authors consideration:
1. Line 212: “We next took advantage of the quantitative nature of SXT contrast and information on the organelles over the entire cell.” I don’t understand the purpose of this single sentence paragraph. The next section doesn’t deal with organelles, but rather virus particles. As far as I am concerned, this sentence could be deleted.
2. Line 378: “These structures resemble multivesicular bodies, a central station in the endocytic-lysosomal pathway required for cytoplasmic transport, maturation, and egress of HSV-1 (ref. 45). The reference does support the idea that HSV-1 uses MVBs for egress, but I don’t think this is a widely-held view amongst those researchers studying HSV-1 egress. Therefore, the authors may want to re-write this sentence.
Author Response
- Line 212: “We next took advantage of the quantitative nature of SXT contrast and information on the organelles over the entire cell.” I don’t understand the purpose of this single-sentence paragraph. The next section doesn’t deal with organelles, but rather virus particles. As far as I am concerned, this sentence could be deleted.
We agree, the sentence has been deleted.
- Line 378: “These structures resemble multivesicular bodies, a central station in the endocytic-lysosomal pathway required for cytoplasmic transport, maturation, and egress of HSV-1 (ref. 45). The reference does support the idea that HSV-1 uses MVBs for egress, but I don’t think this is a widely-held view amongst those researchers studying HSV-1 egress. Therefore, the authors may want to rewrite this sentence.
Thank you for pointing this out. The sentence has been modified as follows:
“These structures resemble multivesicular bodies, essential intermediates in the endolysosomal pathway. It has been suggested that multivesicular bodies most likely have a role in the HSV-1 gB protein-mediated envelopment of egressing viral capsids45”.